# A hotspot of *Toxoplasma gondii* Africa 1 lineage in Benin: How new genotypes from West Africa contribute to understand the parasite genetic diversity worldwide

**Azra Hamidović**[1]*, **Jonas Raoul Etougbétché**[2], **Arétas Babatounde Nounnagnon Tonouhewa**[3], **Lokman Galal**[1], **Gauthier Dobigny**[2,4], **Gualbert Houémènou**[2], **Honoré Da Zoclanclounon**[5], **Richard Amagbégnon**[6], **Anatole Laleye**[5], **Nadine Fievet**[7], **Sylvain Piry**[4], **Karine Berthier**[4], **Hilda Fátima Jesus Pena**[8], **Marie-Laure Dardé**[1,9], **Aurélien Mercier**[1,9]

1 INSERM, Univ. Limoges, CHU Limoges, IRD, U1094, Tropical Neuroepidemiology, Institute of Epidemiology and Tropical Neurology, GEIST, Limoges, France, 2 UAC, EPAC, Laboratoire de Recherche en Biologie Appliquée, Unité de Recherche sur les Invasions Biologiques, Cotonou, Benin, 3 Communicable Disease Research Unit (URMaT), Université d'Abomey-Calavi, Cotonou, Benin, 4 Centre de Biologie pour la Gestion des Populations, IRD, CIRAD, INRA, Montpellier SupAgro, MUSE, Montpellier, France, 5 Laboratoire d'Expérimentation Animale, Unité de Biologie Humaine, Faculté des Sciences de la Santé, Université d'Abomey-Calavi, Cotonou, Benin, 6 Laboratoire de biologie médicale, Centre Hospitalo-Universitaire de la Mère et de l'Enfant Lagune (CHU-MEL), Cotonou, Bénin, 7 UMR216-MERIT, IRD, Université Paris-5, Sorbonne Paris Cité, Paris, France; Centre d'Etude et de Recherche sur le Paludisme Associé à la Grossesse et à l'Enfance (CERPAGE), Cotonou, Bénin, 8 Department of Preventive Veterinary Medicine and Animal Health, Faculty of Veterinary Medicine, University of São Paulo, São Paulo, Brazil, 9 Centre National de Référence Toxoplasmose/*Toxoplasma* Biological Resource Center, CHU Limoges, Limoges, France

* azra.hamidovic@unilim.fr

## Abstract

Through international trades, Europe, Africa and South America share a long history of exchanges, potentially of pathogens. We used the worldwide parasite *Toxoplasma gondii* to test the hypothesis of a historical influence on pathogen genetic diversity in Benin, a West African country with a longstanding sea trade history. In Africa, *T. gondii* spatial structure is still non-uniformly studied and very few articles have reported strain genetic diversity in fauna and clinical forms of human toxoplasmosis so far, even in African diaspora. Sera from 758 domestic animals (mainly poultry) in two coastal areas (Cotonou and Ouidah) and two inland areas (Parakou and Natitingou) were tested for *T. gondii* antibodies using a Modified Agglutination Test (MAT). The hearts and brains of 69 seropositive animals were collected for parasite isolation in a mouse bioassay. Forty-five strains were obtained and 39 genotypes could be described via 15-microsatellite genotyping, with a predominance of the autochthonous African lineage Africa 1 (36/39). The remaining genotypes were Africa 4 variant TUB2 (1/39) and two identical isolates (clone) of Type III (2/39). No difference in terms of genotype distribution between inland and coastal sampling sites was found. In particular, contrarily to what has been described in Senegal, no type II (mostly present in Europe) was isolated in poultry from coastal cities. This result seems to refute a possible role of European maritime trade in Benin despite it was one of the most important hubs during the slave trade

**Data Availability Statement:** All relevant data are within the manuscript and its Supporting Information files.

**Funding:** This work was supported by funds from the French Agence Nationale de la Recherche (ANR project IntroTox 17-CE35-0004 given to AM, funder for LG salary), the region of Nouvelle Aquitaine (funder for AH salary). The funders had no role in study design, data collection and analysis, decision to publish, or preparation of the manuscript.

**Competing interests:** The authors have declared that no competing interests exist.

period. However, the presence of the Africa 1 genotype in Brazil, predominant in Benin, and genetic analyses suggest that the triangular trade was a route for the intercontinental dissemination of genetic strains from Africa to South America. This supports the possibility of contamination in humans and animals with potentially imported virulent strains.

## Author summary

The parasite *Toxoplasma gondii* is a worldwide-distributed pathogen, able to infect all warm-blooded animals. There are important differences in the clinical expression of the infection in direct relation with the parasite genetic profile. In some regions, the geographical structuration of its genetic diversity points towards a crucial role of human activities in some lineages introduction or sorting. Benin is a West African country with a history of extensive transcontinental exchanges. Our genetic study of *Toxoplasma* in Benin shows a surprisingly homogeneous and autochthonous diversity, which contrasts with previous studies from other West and Central African countries. In Benin, the absence of European *Toxoplasma* lineages may be explained by the extreme rarity of the house mouse (*Mus musculus*), a host species that was previously described as highly susceptible to the mouse-virulent African strains. Might Benin be the origin region for the Africa 1 lineage, our results suggest that Guinean Gulf coasts may be a starting point of this lineage towards South America, especially Brazil, during the slave trade. As a whole, the present study provides further insights into the recent evolutionary history of *Toxoplasma gondii* and its consequences on human and animal health.

## Introduction

The protozoan *Toxoplasma gondii* is an obligate intracellular parasite responsible for toxoplasmosis. Its life cycle is complex and involves multiple hosts. Felids, as definitive hosts, excrete the parasite oocysts produced after sexual reproduction in their digestive tract [1,2]. Intermediate hosts (all warm-blooded animals) may then be infected by ingesting oocysts present in the environment [3]. Key reservoir species include rodents, which are felids preferred preys. Other ground-foraging animals, such as poultry, are good bioindicators of *T. gondii* circulation in the environment [4].

In humans, the infection source varies depending on hygiene and eating habits [2]. With an estimated 25% of the world population being infected asymptomatically [5], *Toxoplasma* infection does not appear to be a public health priority, especially on tropical continents such as Africa where much more morbid infections like malaria or tuberculosis are widespread [6]. Yet, toxoplasmosis may be lethal in some large at-risk groups like fetuses or immunocompromised patients [3], but also in immunocompetent patients with severe forms of toxoplasmosis mainly described in South America [7,8]. Some investigations have shown that a link exists between the parasite genotype and the virulence in both mouse [9] and human [5,6,10]. Though the underlying processes remain unknown, such a relationship enhances the importance of documenting *T. gondii* genetic diversity and structure. In sub-Saharan Africa, data on both the genetic diversity of *T. gondii* and the clinical forms associated with this zoonotic disease are scarce [11–14]. However, in West and Central Africa, it was shown that ocular toxoplasmosis could be an important health issue [15] and that cerebral toxoplasmosis was found to be the second cause of death after tuberculosis among HIV-positive patients [10]. In

addition, a recent study conducted in France has described cases of severe toxoplasmosis imported from tropical Africa in immunocompetent patients and associated with strains belonging to the Africa 1 lineage [16]. Therefore, further data on the diversity of *Toxoplasma* from this continent are required to better understand the eco-evolution of the pathogen and the risks for human health.

The Republic of Benin, formerly Dahomey, is located in West Africa, surrounded by Togo, Nigeria, Burkina Faso and Niger. Currently, Benin has an important maritime commerce through its International Sea port of Cotonou [17] with importation of vehicles, rice, fuels and palm oil (https://www.tresor.economie.gouv.fr/Pays/BJ/commerce-exterieur-du-benin) and exportation of raw cotton (34.5% of exportations), coconuts, Brazil nuts and cashews and gold to, inter alia, India, Bangladesh, China, United Arab Emirates and Vietnam (oec.world/en/profile/country/ben/).

During the 17th century, the region was referred to as the Slave Coast [18]. Benin was a major source of African slaves during the so-called Atlantic triangular trade: raw cotton was shipped from America to Europe where it was manufactured and then sent to Africa. There, it was traded for slaves who were deported to Americas (an estimated two to three millions of people were deported from the Bight of Benin; [19,20]) (http://revealinghistories.org.uk/africa-the-arrival-of-europeans-and-the-transatlantic-slave-trade/articles/commerce-and-collecting.html).

Recent evidence has shown that *T. gondii* was already present in Africa 2,000 years ago [21]. However, such extensive and long-term intercontinental exchanges may have fueled the dissemination of many organisms to and from coastal Africa, and this could include pathogens such as non-autochtonous lineages of *T. gondii* through the accidental (e.g. rats) or deliberate (e.g. cats) importation of their hosts from Europe [4,22]. Interestingly, invasive species introduced into Africa may have long remained along the coasts for a while before being disseminated further inland due to late development of inland transports (e.g. rats and mice in Senegal [23,24]; rats along the Benin-Niger corridor [25]). It is thus feasible that this past maritime trade have shaped differences in genetic diversity and population structure of *T. gondii* between continents, but also between coastal and inland areas within Africa. In such context, the main objective of the present study is to map *T. gondii* genetic diversity across diverse regions of Benin and to compare the observed diversity to that found elsewhere in Africa but also Brazil, final destination of millions of slaves for three centuries.

## Methods

### Ethics statements

The Ethical Committee of Research of the Institut des Sciences Biomédicales Appliquées (ISBA), Cotonou, authorized this research in Benin (N°112 the 20/08/18). All experimental procedures were conducted according to European guidelines for animal care ("Journal Officiel des Communautés Européennes", L358, December 18, 1986).

In agreement with the equitable sharing of genetic resources between Benin and France, a Nagoya Protocol on Access and Benefit Sharing was also accepted by the focal point in Cotonou.

### Sampling sites

Four cities of Benin were sampled from May to October 2018. In the south, two coastal cities were selected: Cotonou (6˚23'25.4"N 2˚23'58.6"E), the actual economic capital with a large international commercial seaport, and Ouidah (6˚22'17.3"N 2˚04'29.4"E), an ancient major seaport for the exportation of slaves and goods to the Americas and Caribbean Islands [19,20].

In the north, two inland cities were investigated: Parakou (9˚20'43.5"N 2˚36'20.3"E), an ancient and still important crossroad between Benin, Nigeria, Burkina Faso and Niger, and Natitingou (10˚17'46.4"N 1˚22'56.2"E), the closest city to the Pendjari National park, one of the last large wildlife refuge in West Africa.

In each sampling sites, multiple neighborhoods and villages were sampled, with the systematic oral authorization of owners, village/neighborhood traditional chiefs and representatives of local administrations.

## Poultry sampling

Sampling was performed on 758 domestic animals, mainly hens (*Gallus domesticus*, n = 746) but also ducks (*Anas platyrhynchos*, n = 10), one guinea fowl (*Numida meleagris*) and one turkey (*Meleagris gallopavo*). Individual (age, sex, place of birth) and environmental (soil type, source of water, GPS coordinates at the household level) information were noted for each animal. They were then identified using a ring, an individual number and a picture. Blood was collected from a wing vein using a syringe with a 25G needle and stored at 4˚C.Samples were then centrifuged to separate the serum from blood cells, and immediately serotested on the field.

## *Toxoplasma* Serological examination

Sera from domestic animals and laboratory mice (see below) were screened for *T. gondii* IgG antibodies using the Modified Agglutination Test (MAT) (Toxoscreen DA kits from Biomérieux) [26] with serial dilutions (1:20, the positivity threshold, 1:40, 1:100 and 1:800) as described in the literature [27].

## Strain isolation protocol

Seropositive animals were bought to the owners and slaughtered. Heart and brain were collected, preserved at 4˚C or on ice and processed at the ISBA laboratory (Cotonou) within the next five days. The isolation protocol on Swiss mice follows the description by Galal and collaborators in 2019 [4]. Cryogenized tissue samples of seropositive mice were then stored in liquid nitrogen at the *T. gondii* Biological Resource Centre (BRC), Limoges, France, (http://www.toxocrb.com) for strain preservation and subsequent sub-inoculation.

## 15-Microsatellite genotyping and genetic analyses

**DNA extraction and qPCR.** Toxoplasma DNA extraction from tissues and biological exudates (mouse brain or ascites and poultry tissue digests) was performed with QIAamp DNA mini kit (Qiagen).

The real-time qPCR targeting the non-coding element rep529 (GenBank accession no. AF146527) was used to test to assess the parasitic load of *Toxoplasma* in digests from domestic animals and tissues from seropositive laboratory mice with an unsuccessful *T. gondii* genotyping.

**Microsatellite genotyping.** A 15 microsatellite (MS) markers multiplex PCR was performed as previously described [28]. The 15 loci used are located on 11 different chromosomes of *T. gondii* genome. Among those, some of them are sparsely polymorphic and rather help typing the strain (TUB2, W35, TgM-A, B18, B17, M33, IV.1 and XI.1) while others are highly polymorphic and enable one to distinguish between strains from a same type (M48, M102, N60, N82, AA, N6 and N83). The Beninese multi-locus genotypes (MLGs) were compared to reference strains described in previous studies (S1 Table) in order to allow the assignment.

**Construction of a Neighbor-Joining tree.** The Brazilian strains (that were published using different genotyping techniques [29–35]) were reanalyzed with the 15-MS genotyping

panel described here above. Then, genetic diversity among Beninese (this study), Senegalese [4,14], Gabonese [11] and Brazilian [29–35] *T. gondii* populations (all animal strains) were explored using the software Populations 1.2.32 (1999, Olivier Langella, CNRS UPR9034 http://bioinformatics.org/~tryphon/populations/). To do so, genetic distances (Cavalli-Sforza and Edwads, Dc (1967)) between unique haplotypes from the same locality were calculated and used to reconstruct a Neighbor-Joining tree. All genotypes retrieved in the present study as well as reference strains used to build this distance-based tree are listed in S1 Table.

**Geographical structure, genotypic and genic diversity.** Spatial and diversity analyses were performed only on Africa 1 genotype samples since they were the only ones to be sufficiently well represented (36/39; see Results). The software QGIS V2.18.15 (qgis.org) was used to map the geographical distribution of the sampling locations of the corresponding genotypes. Various diversity indices (Shannon-Wiener diversity index H; Stoddart and Taylor's index G; Simpson's index lambda; Evenness E.5) were calculated on R version 3.4.3 using a Poppr function called "diversity_ci". For these analyses, all strains, even epidemiological clones, were included.

For genic diversity, epidemiological clones of a same GPS point were excluded of this analysis in order to prevent biased clustering. To assess the population structure of *T. gondii* in Benin, as well as to explore potential geographic exchanges and flows, genic differentiation analysis for each pair of population within Benin (namely Cotonou, Ouidah, Parakou and Natitingou) as well as trans-continentally (i.e. between Benin, Senegal, Gabon and Brazil) was performed via Genepop (https://kimura.univ-montp2.fr/~rousset/Genepop.htm) with 100 batches and 5,000 iterations per batch as Markov chain parameters. A p-value for each population pair across all polymorphic loci (M48, M102, N60, N82, AA, N6 and N83) was calculated with Fisher's method.

**Minimum spanning network.** A minimum spanning network using Poppr package on R version 3.4.3 was drawn to visualize the relationships between Africa 1 genotypes from Benin, Senegal, Gabon and Brazil based on Bruvo's genetic distance [36].

## Mouse virulence

We took advantage of the four weeks long isolation protocol (see above) to investigate parasite virulence. To do so, mouse survival was monitored within the four weeks in order to determine *a posteriori* the association between *Toxoplasma* genotype and infection phenotype in mice [22]. Doses of parasites initially present in inocula were arranged into three classes (>25, [25–100[, ≥100 parasites) with a qPCR-based estimation of parasites within digest samples (S1 Table). Once again, only *Africa 1* strains from Benin were included in this particular analysis due to the number of samples and data available. The relation between survival time and dose was studied using a Cox proportional-hazard regression analysis and a Kaplan-Meier method [11] implemented in IMB SPSS Statistics 22.0.

## Results

### Sero-epidemiology of *T. gondii* in poultry

A total of 758 animals were sampled and led to an overall seroprevalence of 15.7% (n = 119/758) (95% IC: 13.1–18.3). Differences with overlapping IC were noted between cities: 21.2% (28/132; 95% IC: 14.24–28.19) in Cotonou, 17.5% (36/206; 95% IC: 12.29–22.66) in Ouidah, 13.9% (29/208; 95% IC: 9.23–18.65) in Parakou and 12.3% (26/212; 95% IC: 7.85–16.68) in Natitingou (26/212).

### Isolation, virulence and genotyping of *T. gondii* isolates from Benin

Out of the 119 seropositive poultry, 69 isolations were attempted and 45 ones were successful. Among the latter ones, 39 could be genotyped using 15-MS genotyping (Fig 1): 36 strains were

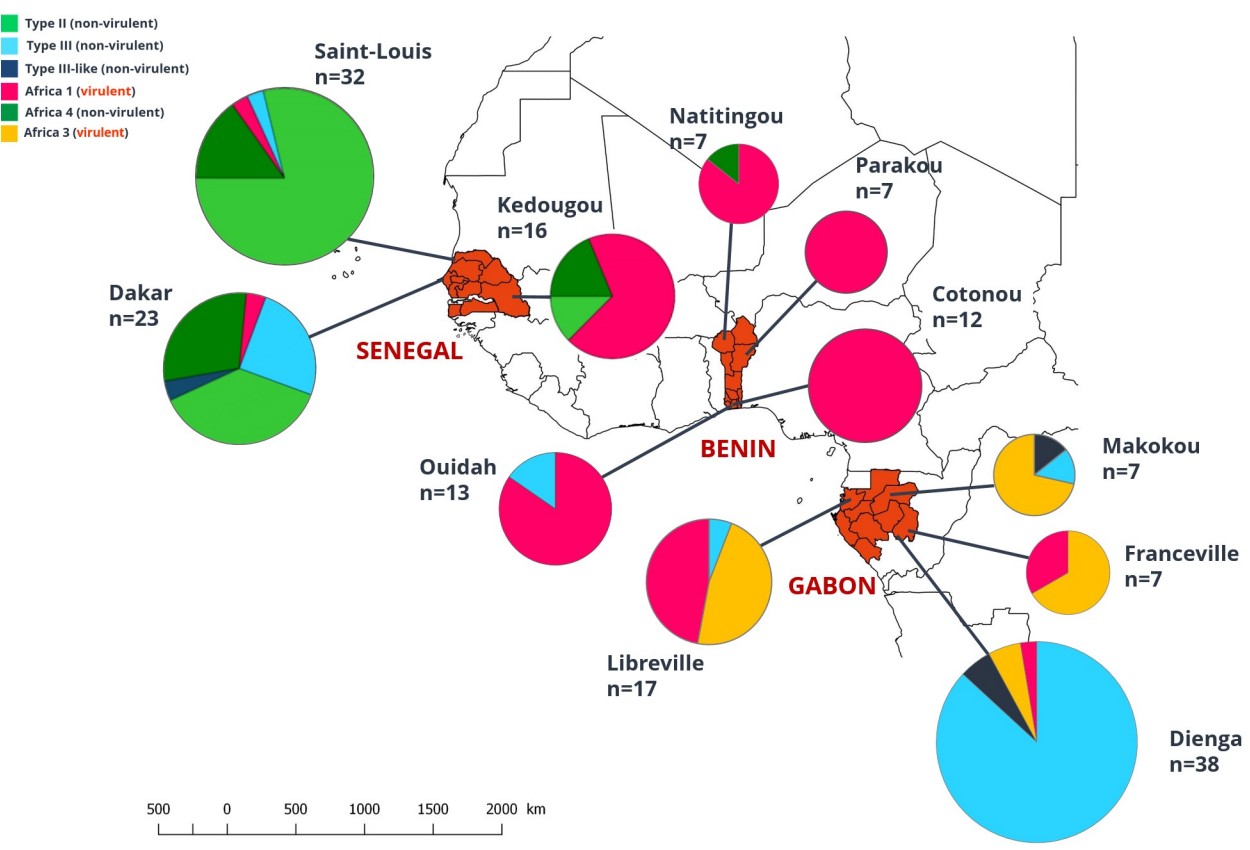

**Fig 1. Geographical distribution of Senegalese (modified from Galal et al., 2019), Beninese and Gabonese *Toxoplasma gondii* strains.** Map of the distribution of *Toxoplasma gondii* of isolates fully genotyped by 15-MS genotyping. Size of pie charts correlate with the total number of isolates and colors indicate the different clonal lineages found.

Africa 1 (the eight typing markers matching with reference strain "FOU"; see S1 Table) with 27 distinct MLGs. They were virulent for laboratory mice with the rapid development of an ascitic fluid usually followed by euthanasia or death after puncture, with 50% of mice being dead at day 10 p.i. (Fig 2). Two others strains from the region of Ouidah were found to be clones of Type III (non-virulent for mice) and produced cysts in the brain of three laboratory mice four weeks p.i. (respectively a total of 80, 128 and 24 cysts per brain).

During the isolation protocol, seven strains lead to an apparently asymptomatic infection, with a positive serology but no brain cyst by direct microscopic examination and unsuccessful genotyping probably due to insufficient DNA amount (positive sample in qPCR but Ct > 34) or unsuccessful sub-inoculation of frozen aliquots back to France. Among those isolation attempts, only one of the inoculated digest could be genotyped and corresponded to Africa 4 variant TUB2 (293pb instead of 291pb) (S1 Table).

## Genetic structure and genotypes geographical distribution

The different genotypes found in Benin and in two other African countries, Senegal [4,14] and Gabon [11] are illustrated in Fig 1. In Benin, Africa 1 is the dominant genotype from South to North.

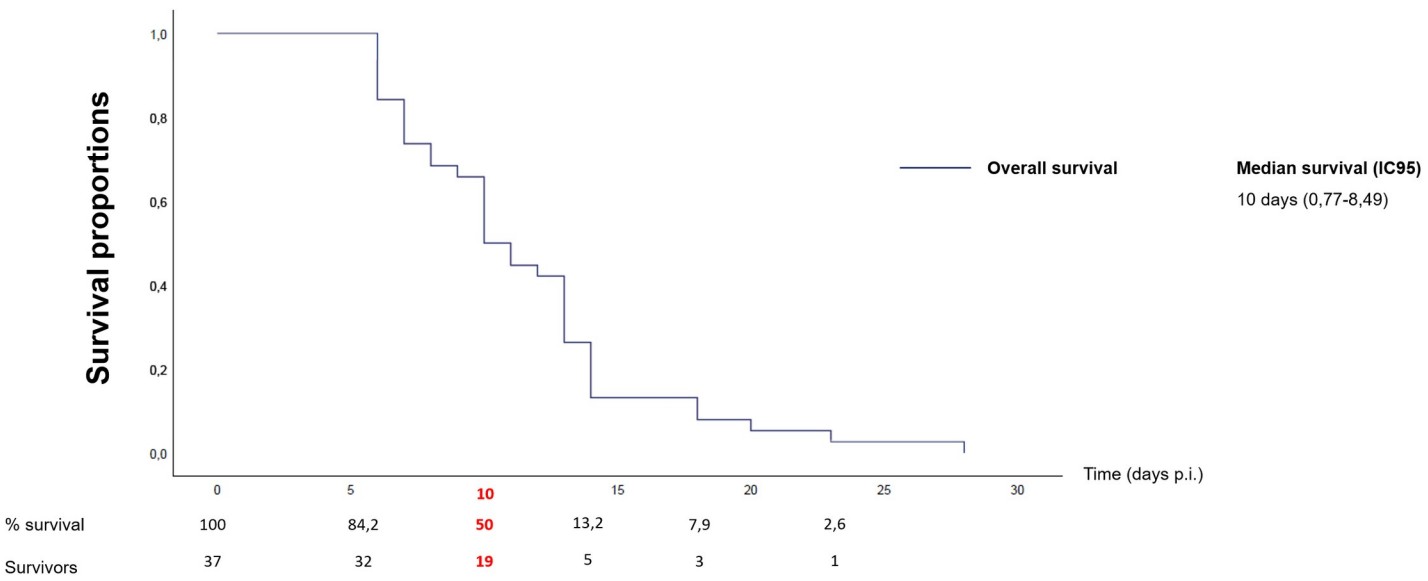

**Fig 2. Survival curve of Swiss mice infected by *Africa 1* strains from Benin.**

The phylogenetic distance-based tree separates the strains from Africa, Europe and South America into three different haplogroups that correspond to Africa 1, Type III and Africa 4 genotypes (Fig 3). No clear structure was retrieved within the Africa 1 lineage. In contrast, the Type III group appears divided into three clusters: one made of only Senegalese genotypes, a second one with only Gabonese genotypes and a third one with the single Type III genotype from Benin together with some strains from Senegal and Gabon.

### Africa 1 population genetics within Benin and comparison with strains from Senegal, Gabon and Brazil

The sample size of Africa 1 strains allowed us to focus on that particular lineage and to perform population genetic analyses on 27 Africa 1 MLGs from Benin, six from Gabon, 12 for Senegal and 19 from Brazil (Table 1). Values point to similar diversity within Senegal and Brazil, a slightly higher diversity in Benin and a lower one in Gabon, where a possibly more clonal population is observed, although this was based on a smaller sampling size.

No significant genetic differentiation was found between pairs of localities within Benin (Table 2). The same analysis performed worldwide (i.e. Benin, Senegal Gabon and Brazil) shows that Benin and Senegal are the only two countries with no significant differentiation (p-value = 0.37) (Table 3) although no obvious common allele was found between each strains of these populations through the MS-genotyping. The minimum spanning network indicates that the Africa 1 strains isolated in Africa and Brazil are closely related (Fig 4); however, no country-specific genetic affinities are observed.

### Discussion

Over the last two decades, the picture of the genetic diversity of *Toxoplasma gondii* in Africa has been slowly drawn thanks to accumulating data and an increasing sampling size [13]. Nevertheless, West Africa remains one of the least documented region of the world in terms of number of isolates and genotypes described. This study significantly contributes to filling such a gap of knowledge with the description of 39 *T. gondii* genotypes from Benin, compared to

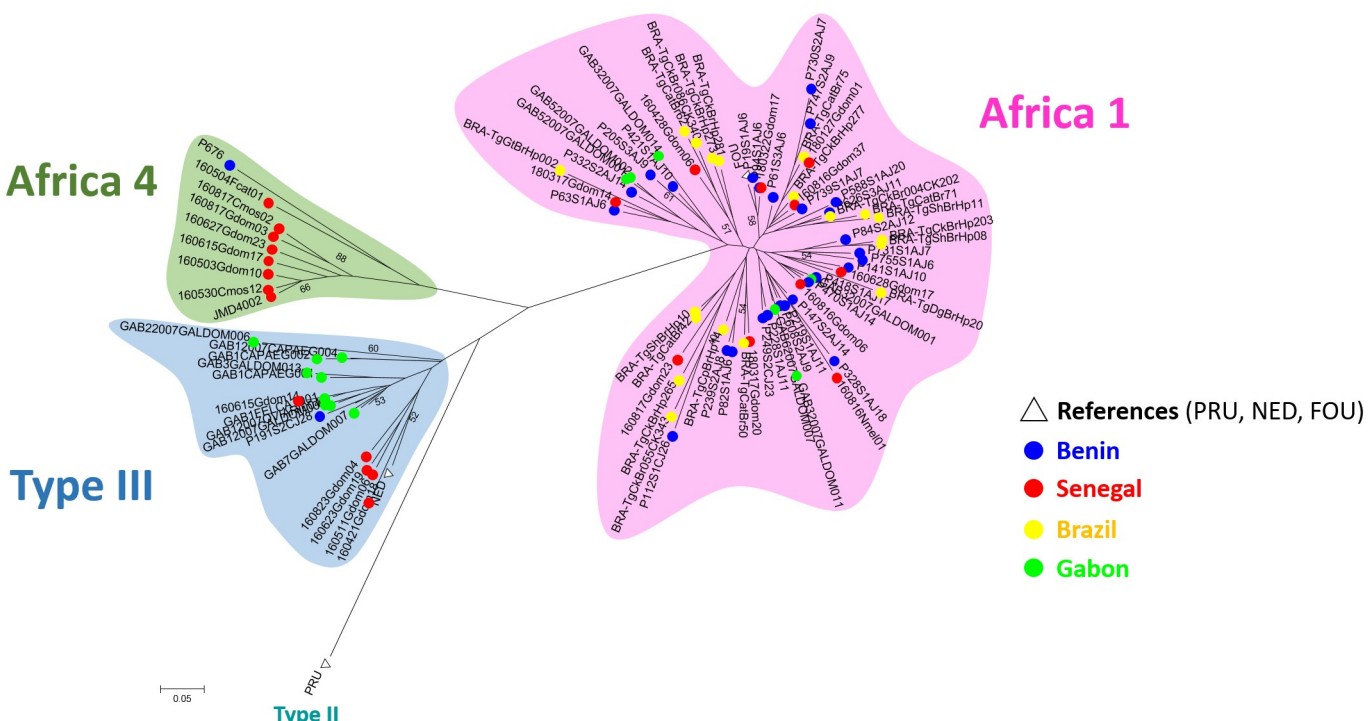

**Fig 3. Global Neighbor-joining tree of genotypes inferred from calculated Cavalli-Sforza distances for 15 microsatellite markers.**

available microsatellite data of *T. gondii* diversity from West (Senegal) and Central Africa (Gabon). It also allows us to investigate the influence of intercontinental exchanges and the potential heritage of the triangular trade on shaping *T. gondii* genetic diversity.

We found three different *T. gondii* lineages that circulate in Benin: Type III, Africa 4 and Africa 1. The Type III clone (2 strains), corresponding to a cosmopolitan lineage already described in various countries on the Africa continent [13], were found in Ouidah, a former Portuguese fort where slave export was once extensive [37]. Type III distribution and circulation remain mysterious, with description of this genotype all around the globe and in both anthropized and isolated areas [4,11,38].

One single Africa 4 sample was successfully genotyped from Boukoumbé, a village at the Togolese border, 80 km east of Natitingou in a remote area. Africa 4 lineage (equivalent to RFLP genotype ToxoDB#20) is a widespread lineage in Africa (Egypt, Ethiopia, Senegal, Mali

**Table 1.** *Toxoplasma gondii* genetic diversity and genotypic diversity.

| Population | N | MLG | eMLG | SE | H | H_rar | G | G_rar | lambda | lambda_rar | E.5 | E.5_rar | Hexp |
|---|---|---|---|---|---|---|---|---|---|---|---|---|---|
| Benin | 36 | 27 | 9,10 | 0,844 | 3,17 | 2,452017 | 20,25 | 11,070675 | 0,951 | 0,9072449 | 0,845 | 0,9361424 | 0,214 |
| Gabon | 8 | 6 | 6,00 | 0,000 | 1,67 | 1,667462 | 4,57 | 4,571429 | 0,781 | 0,7812500 | 0,831 | 0,8308155 | 0,187 |
| Senegal | 15 | 12 | 8,33 | 0,826 | 2,34 | 2,282658 | 8,33 | 8,097062 | 0,880 | 0,8749082 | 0,783 | 0,8006349 | 0,194 |
| Brazil | 26 | 19 | 8,69 | 0,935 | 2,81 | 2,377141 | 14,08 | 10,059880 | 0,929 | 0,8975714 | 0,837 | 0,9124291 | 0,248 |

Genetic diversity and genotypic diversity of Africa 1 estimated for Benin, Gabon, Senegal and Brazil. (N: number of genotypes; MLG: multilocus genotypes; eMLG: expected MLG based on rarefaction; SE: standard error from rarefaction; H: Shannon-Wiener diversity index; H_rar: Shannon-Weimer diversity index after rarefaction; G: Stoddard and Taylor's index; G_rar: after rarefaction; Lambda: Simpson's index; Lambda_rar: Simpson's index after rarefaction; E.5: evenness; E.5_rar: evenness after rarefaction; Hexp: Nei's gene diversity (expected heterozygosity))

**Table 2. P-value for each population pair of Beninese regions across all loci (Fisher's method).**

| Population pair | Chi2 | df | P-value |
|---|---|---|---|
| Cotonou & Ouidah | 8.22146 | 12 | 0.767593 |
| Cotonou & Parakou | 10.139427 | 12 | 0.603730 |
| Ouidah & Parakou | 10.208334 | 12 | 0.597689 |
| Cotonou & Natitingou | 5.630950 | 12 | 0.933532 |
| Ouidah & Natitingou | 5.659725 | 12 | 0.932255 |
| Parakou & Natitingou | 3.100918 | 10 | 0.978948 |

and Gambia), and was repeatedly found in Asia (China, Sri Lanka, United Arab Emirates) [13]. However, the strain isolated in this study was an Africa 4 variant TUB2 (equivalent to ToxoDB#137 [39]), which is a quite uncommon variant of this lineage since it was only described in three African countries (Senegal, Ghana and Mali) [4,13]. This sample was directly genotyped from a digest since the strain isolation "failed": no clinical virulence in laboratory mice and no cerebral cyst were observed despite the positive serological test as well as the positive qPCR reaction on the brain. However, the low DNA amount detected in mouse brains (Ct = 38.37) could explain the genotyping failure. This atypical pattern has already been observed in a previous study on Senegalese strains belonging to the Africa 4 lineage. In our study, four other isolation attempts within the same village (Boukoumbé), without successful genotyping, also showed a similar isolation pattern. Given the geographical proximity and the sharing of this phenotypic trait, it is highly probable that these strains also belong to the Africa 4 lineage. These observations may highlight a bias in the widely used *T. gondii* isolation protocol. The use of a mouse model could disallow the isolation and/or characterization of certain strain genotypes by selecting a part of the diversity inducing a lower DNA amount in the host brain (seemingly, the case for the Africa 4 lineage). The use of knocked-out immunosuppressed mice [40] may solve such a problem but brings ethical concerns and other technics like cell culture are to optimize. However, these two methods would be complicated to implement without proper infrastructures to maintain sterile conditions, which are usually lacking in tropical regions. Considering these limits, our protocol remains the reference technique for now.

Aside from these two lineages, Africa 1 strains were the most widespread (36/39 characterized strains) and were found all over Benin. This lineage is predominant in tropical Africa since it was found in 11 countries (from Guinea to Democratic Republic of the Congo and more sparsely in East Africa) [13] and is regularly detected in France among imported cases in African patients with symptomatic toxoplasmosis by the National Reference Centre (CNR) for Toxoplasmosis [16]. Interestingly, two Africa 1 strains from Cotonou were respectively identical for the 15 MS to two other strains isolated from a pregnant woman and an immunocompromised patient in Denmark [41]. Although the use of microsatellites may lead to some

**Table 3. P-value for each population pair of countries across all loci (Fisher's method).**

| Population pair | Chi2 | df | P-value |
|---|---|---|---|
| Benin & Senegal | 15.06522 | 14 | 0.373711 |
| Benin & Gabon | 26.702961 | 12 | 0.008525 |
| Senegal & Gabon | 21.169324 | 12 | 0.047955 |
| Benin & Brazil | 38.402169 | 12 | 0.000132 |
| Senegal & Brazil | 27.248545 | 14 | 0.017870 |
| Gabon & Brazil | >61.870083 | 12 | <1.03e-008 |

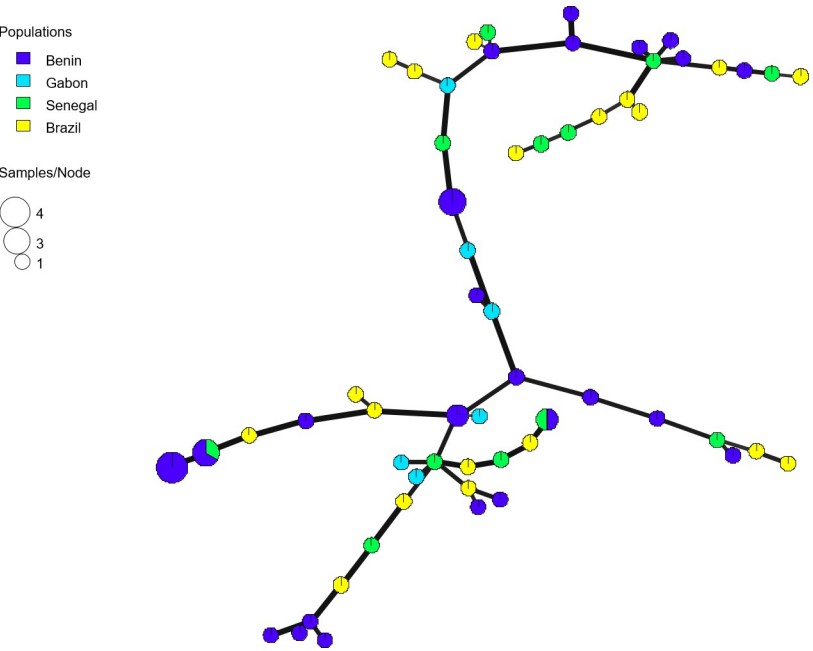

**Fig 4. Minimum spanning network (MSN) showing the relationships between multilocus genotypes (MLGs) of** ***Africa 1*** **lineage from Benin, Senegal, Gabon and Brazil.** Thick and dark lines show MLGs that are more closely related to each other.

homoplasy, such a risk is probably limited here by the number of markers used, especially since our 15 loci are distributed along 11 out of the 14 *Toxoplasma* chromosomes [28]. Therefore, these patients were probably imported cases (infection in Africa), but due to the absence of associated data, we cannot confirm that these patients were from, or travelled to Benin.

In Benin, no genic differentiation between the Africa 1 populations from the four sampled regions was observed, which may support a recent dissemination of this lineage across the country, or reflect an extensive circulation of strains between the four Beninese regions. Its nearly exclusive occurence in all sampled regions is in great contrast with what was previously observed in Senegal [4] and Gabon [11] where two to five lineages are co-occurring in most regions (Fig 2). However, it has to be noted that port regions, the most likely regions in which introduced lineages are expected to be found, were not thoroughly sampled in Gabon as it is the case in Senegal and Benin. So how such different diversity patterns between these three countries could be explained?

First, the absence of the European Type II in Benin and notably on the coast, in contrast to the hypothesis of an introduction of this lineage during the triangular trade in Senegal [4], is surprising since the two countries had a rather similar precolonial and colonial history with Europe. Benin may have had an even more prominent role in the triangular slave trade, with nearly two million slaves leaving the Bight of Benin from 1501–1867 (and more than one million just from Ouidah) while "only" 145.000 slaves were deported from Saint-Louis in Senegal [19]. If the difference is not in the route of introduction of type II strains, the explanations could be related to different factors enabling the persistence of this lineage in these new territories.

One difference between Senegal and Benin is the climate [42]: Senegal is characterized by a typically Sahelian (in the north) and a Sudanian climate (in the south) while Benin has a more tropical climate, from tree savannas (in the north) to subequatorial rain regime (in the south). The possible relationships between this marked climatic demarcation and the *Toxoplasma*

genotypes diversity has already been discussed [13]. In this article, Galal and collaborators proposed that tropical Africa has been isolated from new hosts for quite a long time due to natural barriers such as the Sahara desert and a dense tropical forest, thus resulting in a singular *T. gondii* diversity within hosts living in the tropical area with autochtonous lineages such as Africa 1 [13]. Highly productive biotopes such as the tropical environment probably allow potential hosts of *T. gondii* to allocate a large amount of energy to the evolution of their resistance to pathogens that are abundant there [43]. This could lead to a general increase in the virulence of pathogens [44], including strains of *Toxoplasma* circulating in that environment. These processes would have allowed the selection of more virulent lineages (as seen with experimental mouse models) such as Africa 1. This could also explain the lack of success of the type II lineage to propagate in such tropical environments [13,22,45] both in Africa such as Benin and Gabon, and in South America due to a lack of adaptation of this lineage to this new environment (host immune system and competition with pre-existing and better-adapted indigenous lineages).

The Africa 1 lineage is highly virulent (if not lethal) in laboratory mice [4,11], but also in wild-derived mice [46]. In Senegal, since their introduction to the coast through past maritime trade, the spatial expansion of invasive rodents, mainly the house mouse *Mus musculus domesticus* and the black rat *Rattus rattus*, was driven by road- and river-mediated commercial exchanges [24,47]. It was found that strains of Africa 1 lineage are today seldom found in the port regions of Senegal where the house mouse is the most abundant commensal rodent [24,47] and where non-virulent lineages (type II, followed by type III and Africa 4) appear to be far more common [4]. The opposite was noticed in the inland region of Kedougou where the house mouse has not disseminated yet (Fig 2). To explain this contrasting pattern in the geographical distribution of *T. gondii* lineages in Senegal, Galal and collaborators [4] proposed that the house mouse could have allowed the introduction of the intercontinental lineages type II and type III in the port regions of Senegal and the maintenance of this lineages in these regions, beside the non-virulent lineage Africa 4. The successful invasion of these regions by the house mouse would have gradually eliminated Africa 1 lineage from these areas due to its marked virulence for this species, which is probably the most important prey of cats in this areas. In Benin, a large small mammal survey could not provide any evidence of house mice with the only exception of Cotonou Seaport [25]. This rodent species has therefore not (yet) influenced *T. gondii* populations in this country. In support to such a hypothesis, a similar match between the relative abundance of house mice and the frequency of virulent (Africa 1 and/or 3) *vs.* non-virulent (Type III) *Toxoplasma* strains was observed in Gabon (Franceville *vs.* Makokou; Fig 2) [48].

At the opposite, the most abundant rodents captured in Benin are the native African *Mastomys natalensis*, beside the invasive black rats *R. rattus* and Norway rats *R. norvegicus* [25]. *Mastomys natalensis* was experimentally shown to be resistant to the virulent strains of Type I [49]; and is therefore probably also resistant to Africa 1 since it shares common virulence alleles with Type I [50]. *Rattus norvegicus* is known to be highly resistant to mouse-virulent *T. gondii* strains [51]. No data about the resistance of *R. rattus* to the different *T. gondii* strains is available, but its importance as a significant reservoir for *T. gondii* is doubtful, given that it is an arboreal species, probably having a limited exposition to the oocysts on the soil. The only study from Africa reporting data about the prevalence of *T. gondii* infection in *R. rattus* was from Senegal, where a very low molecular prevalence was found [14].

The composition of the host community, particularly the absence of *M. m. domesticus* from Beninese territory outside the port of Cotonou, together with the probable resistance of these hosts may explain the absence of type II lineage and the Africa 1 lineage predominance in Benin. Our results would therefore support the hypothesis of an important role of *T. gondii* hosts in giving a selective advantage to some *T. gondii* strains [52,53], shaping its population structure [4,14]. This would mainly concern hosts that actually play an important role in

maintaining the transmission cycle of *T. gondii*, namely rodents. These small mammals are indeed the main prey of the domestic cat and are considered the main reservoir of *T. gondii* in the domestic environment. If true, one would expect that the *Toxoplasma* genetic diversity observed in 2018 in Benin will change in coming decades since the dissemination of invasive rodent species (especially the house mouse) is most probably an ongoing process in Benin [25]. Studies on the comparative virulence of different *T. gondii* strains on native African and invasive rodents should be helpful to test this hypothesis.

Although Africa 1 lineage has mainly been described throughout the tropical zone of the African continent (West and Central Africa) with numerous strains, it has also been reported more anecdotally in Brazil (equivalent to PCR-RFLP haplogroup 6) [29–35,39]. In previous studies using fewer markers, genetic proximity had already been found between some of these strains from the two continents and the authors hypothesized a role of the transatlantic slave trade in the 18th and 19th centuries [12,13]. In our study, despite a significant genic differentiation between certain geographical populations of Africa 1 lineage, other analyses (Neighbor-Joining tree, minimum spanning network) showed no genetic differentiation between strains of this lineage. It is expectable that, since the collapse of the trade activities linking Africa to South America by the end of the transatlantic trade, geographical populations of this lineage may have gradually diverged due to their isolation. In a former study, Su and collaborators also came to that conclusion using PCR-RFLP approach by classifying Africa 1 equivalent strains from Brazil mainly in the haplogroup 6, such as FOU (the Africa 1 reference), but also in the haplogroup 14 with an Africa 3 equivalent [39]. However, their analyses found a moderate genetic separation between these two groups, probably influenced only by the geographical separation. Unfortunately, our microsatellites analyses did not allow us to clearly certify the origin of Brazilian Africa 1 strains. Despite that, the wide distribution of this lineage in Africa (from Senegal to Uganda) and its high prevalence in most countries of West and Central Africa is supportive of an introduction starting from Africa to South America, as only limited populations of this lineage were found only in Brazil (*T. gondii* diversity have been thoroughly explored in South America) [13,45]. The inference of the Africa 1 evolutive history will probably require the use of more informative methods, such as high throughput genomics to build phylogenies based on complete genomes of these strains, which has not been done yet with *T. gondii* due to the lack of Africa 1 genomes published from these two regions [54].

As with many zoonoses, exploration of the wild cycle for a global understanding of a pathogen epidemiology should be considered, like in the Pendjari Park (Benin), but remains difficult on safety grounds. For *T. gondii*, except for a few single strains [4], no data exist for African wildlife and characterizing this *T. gondii* diversity will be one of the challenges in the coming years. Nevertheless, with this work, we point to the environmental differences potentially explaining the diversity observed in Benin, Senegal and Gabon.

Previous studies made it clear that port areas and human mediated activities seem to play a crucial role in shaping the epidemiology for a large number of zoonotic diseases such as toxoplasmosis [4,12,22]. These issues raise health challenges for the future in our globalized trading world. In this context, it remains important to continue studying *T. gondii* genetic diversity in the world and especially in Africa where data is still fragmented, particularly concerning the clinical aspects of toxoplasmosis in relation to its genetic diversity.

## Supporting information

**S1 Table. Genotyping results of all *Toxoplasma gondii* strains from Benin, and strains from Gabon, Senegal and Brazil, in addition to reference strains (FOU, NED, PRU).** (XLSX)

## Acknowledgments

We would like to thank Florent Engelmann, the director of the Institut de Recherche pour le Développement (IRD) de Cotonou and its staff for their collaboration on the field mission in Benin. We also thank Dorothée Kindé-Gazard for allowing use to use her laboratory for the isolation protocol at the Faculté des Sciences de la Santé in Cotonou, and Melkior Kouchade, superior officer of Water, Forests and Hunting and Nagoya focal point in Benin for meeting and helping us build the Protocol on Access and Benefit sharing. Finally, yet importantly, we are incredibly grateful to the poultry owners in Benin for accepting to participate to this study.

## Author Contributions

**Conceptualization:** Azra Hamidović, Lokman Galal, Marie-Laure Dardé, Aurélien Mercier.

**Data curation:** Azra Hamidović, Lokman Galal, Marie-Laure Dardé, Aurélien Mercier.

**Formal analysis:** Azra Hamidović, Lokman Galal, Aurélien Mercier.

**Funding acquisition:** Lokman Galal, Aurélien Mercier.

**Investigation:** Azra Hamidović, Jonas Raoul Etougbétché, Arétas Babatounde Nounnagnon Tonouhewa, Honoré Da Zoclanclounon, Richard Amagbégnon.

**Methodology:** Azra Hamidović, Sylvain Piry, Karine Berthier, Marie-Laure Dardé, Aurélien Mercier.

**Project administration:** Aurélien Mercier.

**Resources:** Hilda Fátima Jesus Pena, Marie-Laure Dardé.

**Software:** Azra Hamidović, Lokman Galal, Sylvain Piry, Karine Berthier.

**Supervision:** Gauthier Dobigny, Gualbert Houémènou, Anatole Laleye, Nadine Fievet, Marie-Laure Dardé, Aurélien Mercier.

**Visualization:** Azra Hamidović, Lokman Galal, Marie-Laure Dardé, Aurélien Mercier.

**Writing – original draft:** Azra Hamidović, Lokman Galal, Marie-Laure Dardé, Aurélien Mercier.

**Writing – review & editing:** Azra Hamidović, Jonas Raoul Etougbétché, Lokman Galal, Gauthier Dobigny, Sylvain Piry, Hilda Fátima Jesus Pena, Marie-Laure Dardé, Aurélien Mercier.

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
