## [Decision Letter · Decision Letter 0]

15 Sep 2020

Dear Mrs Hamidovic,

Thank you very much for submitting your manuscript "An Africa 1 hotspot in Benin: How new Toxoplasma gondii genotypes from West Africa contribute to understand the parasite genetic diversity worldwide" for consideration at PLOS Neglected Tropical Diseases. As with all papers reviewed by the journal, your manuscript was reviewed by members of the editorial board and by several independent reviewers. The reviewers appreciated the attention to an important topic. Based on the reviews, we are likely to accept this manuscript for publication, providing that you modify the manuscript according to the review recommendations. 

Sincerely,

Claudia Munoz-Zanzi

Associate Editor

Pikka Jokelainen

Deputy Editor

Reviewer's Responses to Questions

**Key Review Criteria Required for Acceptance?**

**Methods**

-Are the objectives of the study clearly articulated with a clear testable hypothesis stated?

-Is the study design appropriate to address the stated objectives?

-Is the population clearly described and appropriate for the hypothesis being tested?

-Is the sample size sufficient to ensure adequate power to address the hypothesis being tested?

-Were correct statistical analysis used to support conclusions?

-Are there concerns about ethical or regulatory requirements being met?

Reviewer #1: yes

Reviewer #2: The Methods in this manuscript look good, and suitable for the proposed study. However, I do have some comments on some sections: 

i) in lines 150 to 151, the authors mention that they collected the hearts and brains of seropositive animals. Why they collected the hearts and not some other muscular tissue that Toxoplasma is known to be found in infected animals?

ii) in section "Mouse virulence" (lines 198 to 206) there is no mention in the analysis of cysts in infected animals though this data is mentioned in the Results section. How the number of cysts was evaluated? How many mice were used per condition? Which mouse strain was used?

**Results**

-Does the analysis presented match the analysis plan?

-Are the results clearly and completely presented?

-Are the figures (Tables, Images) of sufficient quality for clarity?

Reviewer #1: yes

Reviewer #2: Results were clearly presented by the authors. 

In the section "Isolation, virulence and genotyping of T gondii isolates from Benin" (lines 214 to 221), the authors mentioned that they evaluated the number of cysts in the brain of laboratory mice. As I mentioned in the "Results" section, there is no mention on how this was evaluated. Here, in the "Results" the authors should comment how many cysts were produced with the strains they isolated. Maybe a table comparing the number of isolated cysts per brain per mice from the different strains.

**Conclusions**

-Are the conclusions supported by the data presented?

-Are the limitations of analysis clearly described?

-Do the authors discuss how these data can be helpful to advance our understanding of the topic under study?

-Is public health relevance addressed?

Reviewer #1: yes

Reviewer #2: The authors make a good discussion of the results. On minor note, the authors suggest that the lack of house mice might be connected to the absence of Africa 1 strain, making the Type II strain more common. This would suggest that Type II strain was not found before in Africa, being brought by European colonisers. Maybe the authors could explore more this part, as well as the connection of African strains with Brazilians ones. The authors mentioned in the Introduction (lines 87-88) that ocular toxoplasmosis was important in West and Central Africa. Would these strains be connected to the Brazilians ones that cause/are important in ocular toxoplasmosis in Brazil?

One final comment is about a phrase in "Introduction" (lines 106-111), where the authors give the impression that there was no Toxoplasma in Africa before European colonisers started to port in the shores of African countries. Is there evidence of this? If so, they should include the reference.

**Editorial and Data Presentation Modifications?**

Reviewer #1: minor revision

Reviewer #2: (No Response)

**Summary and General Comments**

Reviewer #1: This is an extension of the concept published earlier in PLOS-Neglected Tropical Diseases from another African country. This is a thorough investigation and I have only minor comments.

1. Line 95 replace nowaday with currently

2. Line 138 replace animals with poultry

3. Line 142- replace insulin needle with—provide dimension

4. Line 150-delete by bleeding

5. PROVIDE a SUPPLENTARY TABLE-giving full details of poultry surveyed-species, source, domestic/wild, antibody titer, adult/juvenile, bioassay details, strain preserved, isolate designation

6. Line 203delete thanks to a qpcr—

7. Line 210—list seroprevalence for each species

8. Line 233-no cyst by direct microscopic examination of brain for tissue cysts

9. Line 313 delete—hardly access----with a remote area

10. Line 316-space before the

11. Line 323—no personal communication-Galalis coauthor

12. Line330—delete but-------optimixe

13. Replace genic with genetic throughout

14. Line 385 need reference for Galal

15. Line 446delete-thanks----

16. Line303-specify extra dozens

Reviewer #2: The manuscript is well written and the authors clearly present their ideas and results. The Discussion is well balanced and the results properly discussed with published literature. Overall, it is a well-sounded manuscript.

PLOS authors have the option to publish the peer review history of their article (what does this mean?). If published, this will include your full peer review and any attached files.

Reviewer #1: No

Reviewer #2: No
---

## [Decision Letter · Decision Letter 1]

12 Nov 2020

Dear Mrs Hamidovic,

We are pleased to inform you that your manuscript 'A hotspot of Toxoplasma gondii Africa 1 lineage in Benin: How new genotypes from West Africa contribute to understand the parasite genetic diversity worldwide' has been provisionally accepted for publication in PLOS Neglected Tropical Diseases.

Best regards,

Claudia Munoz-Zanzi

Associate Editor

Pikka Jokelainen

Deputy Editor

Reviewer's Responses to Questions

**Key Review Criteria Required for Acceptance?**

**Methods**

-Are the objectives of the study clearly articulated with a clear testable hypothesis stated?

-Is the study design appropriate to address the stated objectives?

-Is the population clearly described and appropriate for the hypothesis being tested?

-Is the sample size sufficient to ensure adequate power to address the hypothesis being tested?

-Were correct statistical analysis used to support conclusions?

-Are there concerns about ethical or regulatory requirements being met?

Reviewer #1: yes

Reviewer #2: Yes

**Results**

-Does the analysis presented match the analysis plan?

-Are the results clearly and completely presented?

-Are the figures (Tables, Images) of sufficient quality for clarity?

Reviewer #1: yes

Reviewer #2: Yes

**Conclusions**

-Are the conclusions supported by the data presented?

-Are the limitations of analysis clearly described?

-Do the authors discuss how these data can be helpful to advance our understanding of the topic under study?

-Is public health relevance addressed?

Reviewer #1: yes

Reviewer #2: Yes

**Editorial and Data Presentation Modifications?**

Reviewer #1: no

Reviewer #2: (No Response)

**Summary and General Comments**

Reviewer #1: The authors have adequately revised the paper

Reviewer #2: The authors have modified the manuscript accordingly to the reviewers' comments and suggestions, improving the quality of the text.

This manuscript is in a good format to be published in this journal.

PLOS authors have the option to publish the peer review history of their article (what does this mean?). If published, this will include your full peer review and any attached files.

Reviewer #1: No

Reviewer #2: No

---

## [Editor Report · Acceptance letter]

6 Feb 2021

Dear Mrs Hamidovic,

We are delighted to inform you that your manuscript, "A hotspot of *Toxoplasma gondii* Africa 1 lineage in Benin: How new genotypes from West Africa contribute to understand the parasite genetic diversity worldwide," has been formally accepted for publication in PLOS Neglected Tropical Diseases.

Best regards,

Shaden Kamhawi

co-Editor-in-Chief

Paul Brindley

co-Editor-in-Chief
